

# A comparative study on third trimester fetal biometric parameters with maternal age

Yogitha Poojari[1,*], Prudhvinath reddy Annapureddy[2], Sharmila Vijayan[3], Vinoth Kumar Kalidoss[4], Yuvaraj Mf[5] and Sankaran Pk[6,*]

[1] All India Institute of Medical Sciences, Mangalagiri, Andhra Pradesh, India
[2] Dept of Radio-Diagnosis, All India Institute of Medical Sciences, Mangalagiri, Andhra Pradesh, India
[3] Dept of Obstetrics and Gynaecology, All India Institute of Medical Sciences, Mangalagiri, Andhra Pradesh, India
[4] Dept of Community and Family Medicine, All India Institute of Medical Sciences, Mangalagiri, Andhra Pradesh, India
[5] Department of Anatomy, Saveetha Medical College and Hospital, Chennai, Tamil nadu, India
[6] Department of Anatomy, All India Institute of Medical Sciences, Mangalagiri, Andhra Pradesh, India
[*] These authors contributed equally to this work.

Corresponding author
Sankaran Pk, drpks@live.com

## ABSTRACT

**Background.** Advanced maternal age is an important parameter associated with increased risk of feto-maternal complications and it is an evolving trend in society for women planning for pregnancy in late ages. However there are no studies done whether advanced maternal age has its effects on expression of growth pattern in the fetus. So this study was done to compare the maternal age with the third trimester fetal biometric parameters.

**Methods.** This study was done in 100 antenatal women and divided into two groups: Group 1: optimal maternal age group between 21–29 years of age and Group 2: advanced maternal age 30 and above. The pre-pregnant maternal weight, gestational age and third trimester fetal biometrics using ultrasound are noted and compared between the groups.

**Results.** The maternal weight gain between the groups was optimal but the third trimester fetal parameters were significantly less in advanced maternal age. The abdominal circumference in optimal age group and head circumference in advanced maternal age group was closer to calculated estimated date of delivery (EDD) and would be specific in calculating the gestational age.

**Conclusions.** Though there is no significant difference in maternal weight gain, there are fetal growth restrictions in advanced maternal age group due to which the third trimester fetal parameters are lesser than the optimal age group. Head circumference would be specific in calculating the estimated date of delivery in advanced maternal age group.

## INTRODUCTION

A full term pregnancy lasts up to around 40 weeks from the first day of mother's last menstrual period (LMP) to the birth of the baby. It is divided into three stages, called trimesters: first trimester, second trimester, and third trimester. Conception to about the 12th week of pregnancy marks the first trimester. The second trimester is 13 to 27 weeks, and the third trimester starts about 28 weeks and lasts until birth. In obstetric practice, antenatal ultrasonography plays a very important role in assessing the fetal growth by determining the gestational age using biparietal diameter (BPD), head circumference (HC), abdominal circumference (AC) and femur length (FL). Hadlock values were commonly being used as reference charts in the ultrasound machine and most widely accepted for biometry measurements to calculate estimated fetal weight (*Lalitha & Rao, 2016*).

Fetal growth is influenced by many factors such as race, socioeconomic status, genetics, geographical location, maternal diseases, and number of babies (*Adiri et al., 2015*). The maternal pre-pregnancy weight is important for fetal growth indicating body mass index (BMI) and parity determines fetal weight and growth (*Kirchengast, 2009*). Studies have been done and showed an association between pre-pregnancy maternal weight and fetal growth with mothers of less pregnancy weight gain linked to smaller second and third trimester placental weight and fetal size especially between 28 to 32 weeks. It is understood that development of fetal body structures such as head, femur and abdomen correlates with the general fetal size and weight (*Favour, Oyakhire & Aigbogun, 2017*). Maternal age is very important parameter because childbirth at a young age (adolescent) and advanced age is associated with increased risk of adverse maternal perinatal outcomes, such as postpartum hemorrhage, eclampsia, and cephalopelvic disproportion, as well as adverse infant outcomes including preterm birth, poor fetal growth, low birth weight, and neonatal mortality (*Khalil et al., 2013*). Thus the advanced maternal age antenatal women needed complete maternal and fetal surveillance for any birth related or fetal complications. Most complications remain independent of important known confounders such as poverty, inadequate prenatal care and/or weight gain during pregnancy (*Cavazos-Rehg et al., 2015*). All these previous findings show an importance of maternal age in general fetal growth pattern and its related complications. However there are lots of studies done to correlate complications related to advanced maternal age but there are no studies to compare the significant fetal growth differences in advanced antenatal women without any complications. If there are significant differences in babies born to advanced maternal age that may have an impact on growth and its well being in the society. So this study was done to compare the third trimester fetal biometric parameters, gestational age and weight gain between the maternal ages.

## MATERIALS & METHODS

This cross sectional study was conducted in 100 third trimester antenatal women 32–34 weeks gestational age attending the Obstetrics O.P.D at the All India Institute of Medical Sciences (AIIMS), Mangalagiri, Andhra Pradesh, India during the period of February 2021 to August 2021 (AIIMS/MG//IEC/2020-21/71). The samples were divided into two groups:

group 1: optimal age group ($n = 77$) maternal age between 20 to 30 years and group 2: advanced age ($n = 23$) with maternal age 30 years and above attending AIIMS, Mangalagiri O.P.D during their routine follow up. The antenatal women with natural conception of pregnancy who had previous regular menstrual cycles and well known last menstrual period (LMP) willing to participate in this study were included. The antenatal women not willing to participate and with any medical and surgical conditions associated with pregnancy like diabetes, hypertension etc., consumption of alcohol, chewing of tobacco, smoking, assisted reproductive techniques were excluded from the study. The details of the study were explained and written consent was obtained. Gestational age, pre-pregnant maternal weight and trans-abdominal ultrasound measurements of head circumference, abdominal circumference, femur length, biparietal diameter and estimated fetal weight were noted and tabulated. If the pre-pregnant weight is not known then first trimester weight is noted as pre-pregnant weight and weight gain is calculated accordingly. The maternal age and fetal parameters were summarized as mean and standard deviation. The student t test was utilized for mean comparisons and categorical variables like maternal education and occupation status were summarized as frequency and proportion. The comparison between maternal age and fetal parameters were analyzed using Pearson or Spearman's correlation based on normality. $P$ value less than 0.05 is taken as significant.

# RESULTS

The mean (SD) age of the antenatal women were 23 (3.1) years and 32.8 (1.8) years in optimal and advanced maternal age group respectively. Most of the study participants were primigravida in both groups 48% and 43%, followed by second gravida (31.2%) in optimal maternal age group and third gravid (34.8%) in advanced maternal age group. This difference does not have statistical significance (Table 1).

## Maternal weight gain

The mean (SD) first trimester weight of study participants was 62.6 (6.6) kg in optimal maternal age group and 58.6 (11.2) kg in advanced maternal age group. The mean (SD) weight gain in both groups were 8.5 (3.5) kg and 8.7 (3.5) kg respectively. The weight gain was not statistically different between the study groups. The mean (SD) gestational age as per LMP of optimal maternal age was 32.6 (1.7) which was statistically ($p$ value = 0.015) lower than the mean (SD) gestational age of advanced maternal age 33.6 (1.6) (Table 1) (Figs. 1 and 2).

## Third trimester fetal parameters and gestational age estimation

The mean (SD) of head circumference, abdomen circumference, femoral length, biparietal diameter and estimated fetal weight were significantly lesser in advanced maternal age group (Table 2). The mean (SD) gestational age as per LMP was 228.2 (12.2) days, as per head circumference 239.5 (12.2) days, as per abdomen circumference 231.8 (13.1) days, as per femoral length 233.7 (12.1) days and as per biparietal diameter was 233.1 (11.8) days in optimal maternal age. Among the advanced maternal age the mean (SD) gestational age as per LMP was 235.3 (11.6) days, as per head circumference 231.7 (12.7) days, as per

**Table 1  Maternal age, parity, weight gain and gestational age between the two groups.** Distribution of basic details of study participants.

| Item | | Maternal age <30 years group $N = 77$ | Maternal age ≥30 years group $N = 23$ | *p*-value |
|---|---|---|---|---|
| Maternal age in years | Mean(SD) | 23.2(3.1) | 31.8(1.8) | <0.001* |
| Birth | 1 | 37(48.1) | 10(43.5) | |
| Order | 2 | 24(31.2) | 5(21.7) | 0.355 |
| N | | | | |
| (%) | ≥3 | 16(20.8) | 8(34.8) | |
| First trimester weight | Mean (SD) in kg | 62.6(6.6) | 58.6(11.2) | 0.074 |
| Weight gain | Mean(SD) in kg | 8.5(3.5) | 8.7(3.5) | 0.789 |
| Gestational age as per LMP | Mean (SD) in weeks | 32.6(1.7) | 33.6(1.6) | 0.015* |

Notes.

An asterisk (*) indicates significant difference.

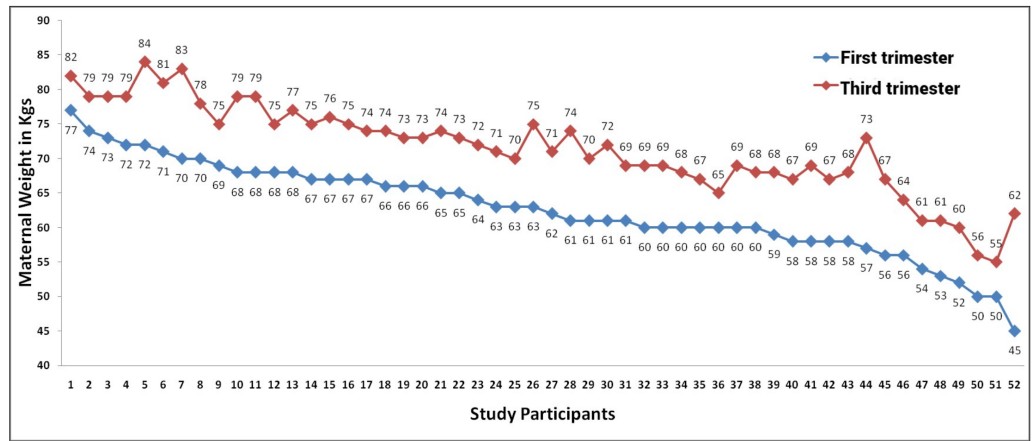

**Figure 1  Maternal weight in first and trimester of study participants in maternal age <30 years group.**

abdomen circumference 223.7 (13.2) days, as per femoral length 226.4 (13.1) days and as per biparietal diameter 225.7 (9.9) days (Table 3). The gestational age as per LMP was statistically higher (*p*-value = 0.016) in advanced maternal age group compared to optimal maternal age group. Whereas the gestational age as per fetal third trimester parameters were statistically lower in advanced maternal age years group.

The median (IQR) difference in gestational age estimation using LMP and as per head circumference, abdomen circumference, femoral length, biparietal diameter was −12.5 (−22.5 to 1.0) days, −3 (−16.5 to 10.5) days, −5 (−19 to 8) days and −7 (−17 to 6) days respectively in maternal age <30 years group. The median (IQR) difference in gestational age estimation using LMP and as per head circumference, abdomen circumference, femoral length, biparietal diameter was 5 (−15 to 17) days, −314 (−1 to 25) days, 10 (−3 to 23) days and 12 (−3 to 20) days respectively in maternal age <30 years group (Table 4). The difference in age estimation using LMP and fetal parameter were significantly different between both groups.

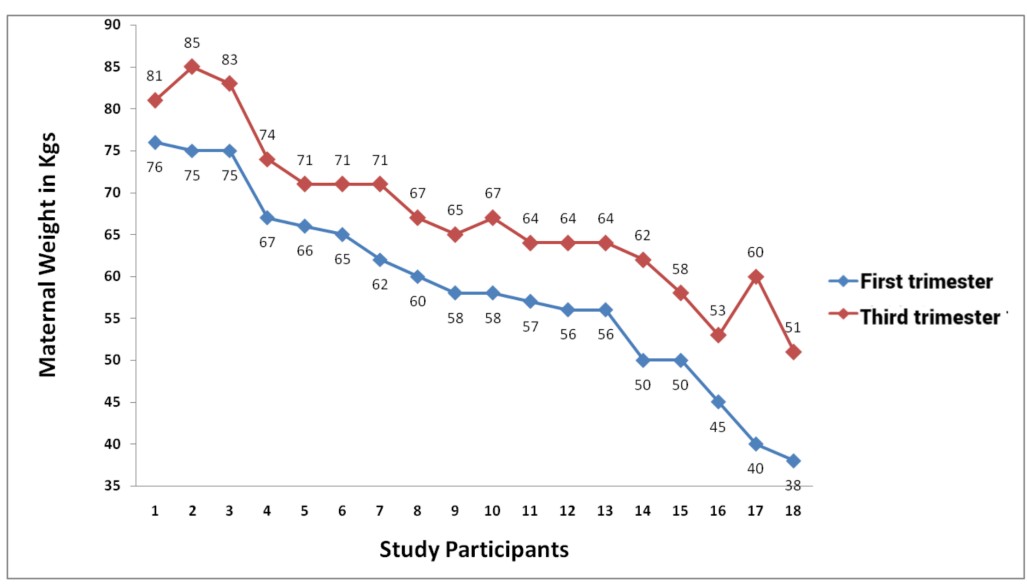

**Figure 2** Maternal weight in first and trimester of study participants in maternal age ≥30 years group.

**Table 2** Comparison of third trimester fetal biometric parameters between the two maternal age groups. Fetal parameter of study participants.

| Fetal parameters | Maternal age <30 years group N = 77 | Maternal age ≥30 years group N = 23 | p-value |
|---|---|---|---|
| Head circumference mean (SD) millimeter | 306.6(12.5) | 298.2(13.6) | 0.007* |
| Abdomen circumference mean (SD) millimeter | 290.4(19.8) | 278(19.7) | 0.017* |
| Femoral length mean (SD) millimeter | 64.6(3.7) | 62.4(4.1) | 0.019* |
| Biparietal diameter mean (SD) millimeter | 82.9(3.9) | 80.4(3.4) | 0.006* |
| Estimated fetal weight mean (SD) gram | 2,191.2(385.8) | 1,944(382.2) | 0.010* |

Notes.
   An asterisk (*) indicates significant difference.

**Table 3** Gestational age estimation using fetal biometric parameters and comparison between maternal age group. Gestational age estimation using various parameters of study participants.

| Gestational age | Maternal age <30 years group N = 77 | Maternal age ≥30 years group N = 23 | p-value |
|---|---|---|---|
| As per LMP mean (SD) days | 228.2(12.2) | 235.3(11.6) | 0.016* |
| As per head circumference mean (SD) days | 239.5(12.2) | 231.7(12.7) | 0.008* |
| As per abdomen circumference mean (SD) days | 231.8(13.1) | 223.7(13.2) | 0.009* |
| As per femoral length mean (SD) days | 233.7(12.1) | 226.4(13.1) | 0.010* |
| As Per Biparietal Diameter Mean (SD) Days | 233.1(11.8) | 225.7(9.9) | 0.015* |

Notes.
   An asterisk (*) indicates significant difference.

## DISCUSSION

Female fertility in the entire reproductive life span differs with ages, during adolescent ages there will be anovulatory cycles followed by peak ovulatory cycles between 20 to 30

**Table 4  Differences between gestational age of EDD and fetal biometry and compared between the maternal ages.** Difference in gestational age estimation of study participants.

| Difference in gestational age | Maternal age <30 years group  N = 77 | Maternal age ≥30 years group  N = 23 | *p*-value |
|---|---|---|---|
| Head circumference median (IQR) days | −12.5(−22.5 to 1.0) | 5(−15 to 17) | <0.001* |
| Abdomen circumference median (IQR) days | −3(−16.5 to 10.5) | 14(−1 to 25) | <0.001* |
| Femoral length median (IQR) days | −5(−19 to 8) | 10(−3 to 23) | <0.001* |
| Biparietal diameter median (IQR) days | −7(−17 to 6) | 12(−3 to 20) | <0.001* |

**Notes.**
An asterisk (*) indicates significant difference.

years and finally declining slowly after thirties (*Balasch, 2010*). The probability of successful pregnancy is low at the beginning and slowly decreases after the increasing age after a period of optimal fertility. There are various studies showing advanced maternal age associated with range of adverse outcomes in pregnancy like low birth weight (*Jolly et al., 2000*), pre-term birth, still birth (*Flenady et al., 2011*), small for gestational age (*Virginia, Howard & Robin, 2001*), macrosomic babies (*Kenny et al., 2013*), increased rates of caesarean section (*Janssens, Wallace & Chang, 2008*), hypertensive syndromes and gestational diabetes (*Hsieh et al., 2010*). Evidences from these previous studies shows the optimal age for child bearing in between 21–29 years of age with high consumption rate and lowest adverse events. There is a trend in recent working women to postpone the pregnancy to mid thirties despite of these facts and evidences had laid importance in finding significant fetal growth pattern in normal pregnancies. This is the first study to analyze the impact of maternal age with the fetal growth patterns using third trimester fetal measurement. Also the fetal growth is associated with maternal somatic parameters like pre-pregnant weight and gestational weight gain have been recorded in this study.

## Fetal biometry and gestational age

The mean gestational age for optimal maternal age group and advanced age group was 32 weeks and 33 weeks respectively (Table 1). There were significant differences in the third trimester fetal biometric measurements between the two maternal age groups (Table 2). The fetal parameters like head circumference, abdominal circumference, femur length, biparietal diameter and fetal weight was significantly more in optimal maternal age group and significantly less in advanced age group. These findings are in accordance with *Metcalfe, Tough & Johnson (2013)* where the mean crown rump length in first trimester was significantly less in advanced maternal age and *Blomberg, Tyrberg & Kjlhede (2014)* where mean birth weight was significantly decreased in advanced maternal age group. The appropriate biological explanation for this less fetal biometry in advanced maternal age may be due to accelerated placental aging and increased oxidative stress (*Lean et al., 2014*) leading to changes in placental vasculature and myometrial spiral arteries leading to decreased blood flow to placenta, uteroplacental under perfusion and placental infarcts making difficult for the older women to adapt the increased demands of pregnancy. Also increasing maternal age is associated with increased placental weight which may be due to compensatory mechanism for placental dysfunction to balance the

reduced uteroplacental blood flow to prevent fetal hypoxia (*Care et al., 2015*; *Torres et al., 2017*). Head circumference of the fetus is very important fetal biometry which indirectly correlates with the brain development. This association of advanced maternal age with significantly less head circumference suggests significant less brain growth and further the child neurodevelopment profile has to be studied. The exact mechanism of different fetal growth pattern in optimal and advanced maternal age group requires additional research. It has been speculated that lower fetal growth pattern in advanced maternal age may be due to less effective nutrient partitioning or transfer of nutrients to the fetus (*Fall et al., 2015*).

## Maternal weight gain

In this study there was weight gain upto 8 kg in both the groups indicating optimal weight gain during the entire pregnancy and there were no significant differences in both the groups (Table 1) (Figs. 1 and 2). The mean first trimester weight was around 62 kg in optimal age group and 58 kg in advanced age group and all the antenatal women were of normal Indian women weight. The optimal weight gain during the entire pregnancy is between 7 to 15 kg if it is more than 15 kg it is excess weight gain leading to macrosomia and preterm birth and if it is less than 7 kg leads to low birth weight and preterm delivery (*Luke & Brown, 2007*). Even though there was no significant difference in the maternal weight gain between the groups the fetal biometry and growth was significantly less in advanced maternal age group extending further to leans about various factors that might lead to uteroplacental insufficiency resulting in these findings. Maternal age is the important risk factor for the pregnancy weight gain which may lead to overweight and this may trigger gestational diabetes mellitus, macrosomia leading to preterm birth and other adverse pregnancy outcomes (*Yang et al., 2015*).

## Estimated gestational age

According to calculated estimated date of delivery (EDD) from last menstrual period (LMP) and the estimated gestational age from various third trimester fetal biometric parameters were compared for any significant differences (Tables 3 and 4). In this study there were significant differences with all third trimester fetal parameters in calculating the gestational age between the groups. The abdominal circumference was very much significant and approximate to LMP calculated EDD in the optimal maternal age and head circumference was very much near to calculated EDD in advanced maternal age group. So the abdominal circumference in optimal age group and head circumference in advanced maternal age group would be specific in calculating the gestational age and to plan for parturition in cases where the LMP is not known. *Karki, Sharmqa & Rauniyar (2006)* conducted study in antenatal women starting from first trimester till the birth and found during first trimester crown rump length (CRL) is the best fetal parameter correlating with the gestational age and in second trimester abdominal circumference best correlating with the gestational age. During third trimester it is the head circumference that correlates best with the gestational age (*Karki, Sharmqa & Rauniyar, 2006*). However that study was not specific to maternal age correlating the fetal gestational age. Considering all these facts advanced maternal age can have complications starting from fertility, first trimester, third trimester, labor,

perinatal mortality, postpartal mortality and chromosomal and genetic disorders of the fetus and appropriate care and diagnostic test must be done at regular period of time to prevent all these complications (*Ales, Druzin & Santini, 1990*).

# CONCLUSION

This study compared the third trimester fetal biometric parameters and weight gain of antenatal women between the maternal age group. This study concludes that there were significant restrictions in fetal growth parameters of advanced maternal age group. This could result in a great impact in the further growth and development of child in the society. Also there were no significant differences in weight gain between the antenatal women of different age groups. Furthermore the abdominal circumference in the optimal maternal age and head circumference in the advanced maternal age were closer to gestational age and to the expected date of delivery. Thus, maternal weight not only determines the fetal growth, maternal age is important in healthy fetal growth.

## Funding
The authors received no funding for this work.

## Competing Interests
The authors declare there are no competing interests.

## Author Contributions
- Yogitha Poojari conceived and designed the experiments, performed the experiments, authored or reviewed drafts of the article, and approved the final draft.
- Prudhvinath reddy Annapureddy conceived and designed the experiments, performed the experiments, analyzed the data, authored or reviewed drafts of the article, and approved the final draft.
- Sharmila Vijayan conceived and designed the experiments, performed the experiments, analyzed the data, prepared figures and/or tables, authored or reviewed drafts of the article, and approved the final draft.
- Vinoth Kumar Kalidoss conceived and designed the experiments, performed the experiments, analyzed the data, prepared figures and/or tables, and approved the final draft.
- Yuvaraj Mf analyzed the data, prepared figures and/or tables, authored or reviewed drafts of the article, and approved the final draft.
- Sankaran Pk conceived and designed the experiments, performed the experiments, prepared figures and/or tables, authored or reviewed drafts of the article, and approved the final draft.

## Human Ethics
The following information was supplied relating to ethical approvals (*i.e.*, approving body and any reference numbers):

This study was approved by the All India Institute of Medical Sciences (AIIMS/MG/IEC/2020-21/71).

## Ethics

The following information was supplied relating to ethical approvals (*i.e.*, approving body and any reference numbers):

This study was approved by the Institutional Ethical Committee (AIIMS/MG/IEC/2020-21/71).

## Data Availability

The raw data are available in the Supplemental Files.

## Supplemental Information

Supplemental information for this article can be found online at http://dx.doi.org/10.7717/peerj.14528#supplemental-information.

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
