# Peer review of "A comparative study on third trimester fetal biometric parameters with maternal age"

_PeerJ, doi:10.7717/peerj.14528_

## Round 0.1 · original submission · Major Revisions

Dear Authors,

Our experts have appreciated the idea and presentation of the manuscript with high clinical significance. However, they have suggested few changes or corrections which may be incorporated or justified.

The major concerns were:

1. The duration of study and selecting only 3rd trimester: please justify the importance.

2. Demographic and other physiological or pathological parameters of the subjects were not provided: please incorporate or justify their exclusion.

3. Periodic weight gain especially the pre pregnancy weight were not provided and periodically the weights were not measured or not shown: any special reason for that?

4. Certain parameters e.g., height, BMI, nutrition, socioeconomic, and other factors which directly affects the fetal parameters were not mentioned.

5. The statistical model used to analyse the results were not mentioned and discussed properly.

6. Keywords should be according to MeSH (Medical Subject Headings) network.
7. The conclusion should be the outcome of your research not the repetition of data collected/analyzed.

8. Discussion should be more impactful.

9. However, the journal gives liberty to choose reference writing pattern in pre publication phase but all the references should be uniform and they should follow any one of standard referencing styles.

Please incorporate all the corrections or justify (if you do not agree with reviewer's comments). Resubmit asap to avoid delay.

Regards and Best of luck.

Reviewer 1 ·

Basic reporting

Basic reporting is adequate. Pls check sentence case & capitalization at places. (Marked in the pdf)

Experimental design

Optimum, however pls indicate statistical method and model clearly as marked in the pdf file.

Validity of the findings

Adequate. Pls write the statistical model/ method for better clarity in material and methods section as also indicated in text.

Additional comments

Pls improve the conclusion , currently is seems a bit loose ended. Specify what you have inferred from your findings.

Annotated reviews are not available for download in order to protect the identity of reviewers who chose to remain anonymous.

·

Basic reporting

1. Some sentences need to be reframed. Grammatical errors observed in few lines.
2. Literature provided is sufficient to support the study findings.
3. manuscript text is structured. Complete data has been shared. However, titles of figures are not places as per PeerJ instructions.
4. Information provided in article is complete.

Experimental design

The design and methodology is explained in detail. the research question is well framed as well as as per need of hour.

Validity of the findings

The results are tabulated appropriately and conclusion is clearly stated at the end of manuscript.

Additional comments

Apart from grammatical corrections and sentence reframing, article needs minor improvements specially adding more weight to discussion section. Keywords should be according to MeSH (Medical Subject Headings) network and can be easily searched at https://meshb.nlm.nih.gov/.
Conclusion in abstract is not correct, they are actually the results.
Titles of figures should be placed below the figure as per Instructions of PeerJ.
In text citations need to be reviewed. Single author studies are also cited in text with 'et.al.' .
The line no. is given below for sentences requiring corrections
38, 92, 93, 104.
Sentences need to be reframed are given below
line no. 60-63, 69-70, 73-76

·

Basic reporting

Some typing mistakes eg.optimal maternal age group 21- 19 years.
Reference are not in proper manner.

Experimental design

Aims and objectives are very good definitely that will help to know fetal growth and for the management of complication in pregnancy.

Validity of the findings

I article should more specific to meet peerj standards.

Article measuring fetal biometrics in two maternal age group in third trimester in pregnency.
1. Study duration - 7 months
Normal term pregnency 37 weeks to 42 weeks. Study duration is less for these research, Less detail about EDD.
2. Article taking measurement only in third trimester and one Key part of article is Pre pregnant Wight of subjects is very doubtful because it is not taken under the author's supervision. So there is chances of error.
3. Including and Excluding criteria are very poor. Article not included hight, BMI, nutrition, socioeconomic, and other factor in this study which directly affects the fetal parameters. Only weight was taken and first trimester weight is not under supervision.
4. No proper history of subjects included in the article which directly affects the results. ( addiction, smoke, diet, religion).
5. Complete development of any animal starts from a single cell whether it is animal or human being. All the phenotypic characteristics are decided by the genome present in this 1st cell. But the morphological features are not taking into consideration in this study

Additional comments

Aims are very good and also helps in management of complicated pregnency in advance maternal age group.

---

## Round 0.2 · accepted · Accept

Dear Dr. Pk,

It is my pleasure to inform you that as per the recommendation of our expert reviewers, the manuscript "A comparative study on Third trimester Fetal biometric parameters with Maternal age" has been Accepted for publication in PeerJ.

This is an editorial acceptance and you will be contacted with the list of further tasks before publication. So, I request you to be available for a few days to make the necessary things asap.

Regards and good luck with your future submissions.

·

Basic reporting

Basic reporting is appropriate

Experimental design

Author has mentioned the type of study design and information provided is relevant to the study conducted.

Validity of the findings

This is a unique study of its kind on feto-maternal parameters.

Additional comments

Practical implications of this study to the clinicians or general public can be elaborated.